Maximum aerobic speed, maximum oxygen consumption, and running spatiotemporal parameters during an incremental test among middle- and long-distance runners and endurance non-running athletes

Casado Arturo arturocasado1500@gmail.com 1
Tuimil José Luis 2
Iglesias Xavier 3
Fernández-del-Olmo Miguel 1
Jiménez-Reyes Pedro 1
Martín-Acero Rafael 2
Rodríguez Ferran A. ferran.a.rodriguez@gmail.com 3
1 Centre for Sport Studies, Universidad Rey Juan Carlos , Fuenlabrada , Madrid , Spain
2 Faculty of Sports Sciences and Physical Education, Universidad de La Coruña , La Coruña , Galicia , Spain
3 INEFC-Barcelona Sports Science Research Group, Institut Nacional d’Educació Física de Catalunya (INEFC), Universitat de Barcelona , Barcelona , Cataluña , Spain
Plavec Davor
Electronic publication date: 2022 Oct 5
Publication date: 2022
Volume: 10
Electronic Location ID: e14035
Received 2022 Jun 9; Accepted 2022 Aug 18
Copyright: ©2022 Casado et al.
Copyright year: 2022
Copyright holder: Casado et al.
License: This is an open access article distributed under the terms of the Creative Commons Attribution License, which permits unrestricted use, distribution, reproduction and adaptation in any medium and for any purpose provided that it is properly attributed. For attribution, the original author(s), title, publication source (PeerJ) and either DOI or URL of the article must be cited.
License URL: https://creativecommons.org/licenses/by/4.0/

Keywords: Maximal oxygen uptake, Performance, Maximal aerobic speed, Running, Spatiotemporal parameters

Funding: The authors received no funding for this work.

==============================
Background

Maximal aerobic speed (MAS) is a useful parameter to assess aerobic capacity and estimate training intensity in middle- and long-distance runners. However, whether middle- and long-distance runners reach different levels of MAS compared to other endurance athletes with similar V̇O2max has not been previously studied. Therefore, we aimed to compare V̇O2max, MAS and spatiotemporal parameters between sub-elite middle- and long-distance runners (n = 6) and endurance non-runners (n = 6). In addition, we aimed to compare the maximal blood lactate concentration [BLa] experienced by participants after conducting these tests.

Methods

Telemetric portable respiratory gas analysis, contact and flight time, and stride length and rate were measured using a 5-m contact platform during an incremental test at a synthetic athletics track. V̇O2, heart rate, respiratory quotient values in any 15 s average period during the test were measured. [BLa] was analyzed after the test . Running spatiotemporal parameters were recorded at the last two steps of each 400 m lap. A coefficient of variation (%CV) was calculated for each spatiotemporal variable in each participant from 8 km h−1 onwards.

Results

Whereas runners reported faster MAS (21.0 vs. 18.2 km h−1) than non-runners (p  =  0.0001, ES = 3.0), no differences were found for V̇O2max and maximum blood lactate concentration during the running tests (p > 0.05). While significant increases in flight time and stride length and frequency (p < 0.001, 0.52 ≤ ηp2 ≤ 0.8) were observed throughout the tests, decreases in contact time (p < 0.001, ηp2=0.9) were reported. Runners displayed a greater %CV (p = 0.015) in stride length than non-runners. We conclude that middle- and long-distance runners can achieve a faster MAS compared to non-running endurance athletes despite exhibiting a similar V̇O2max. This superior performance may be associated to a greater mechanical efficiency. Overall, runners displayed a greater ability to modify stride length to achieve fast speeds, which may be related to a more mechanically efficient pattern of spatiotemporal parameters than non-runners.

Introduction

Maximal aerobic speed (MAS) is considered a useful parameter to predict performance, test training progression and prescribe and program training in middle- and long-distance runners (Billat et al., 1994b; McLaughlin et al., 2010; Saunders et al., 2004). MAS is defined as the minimum running speed necessary to attain maximal oxygen uptake (vV̇O2max) during an incremental treadmill running test (Billat et al., 1994b; Billat et al., 1994a; di Prampero et al., 1986). MAS can also be estimated in the field by employing an incremental track running test (Leger & Boucher, 1980).

On the other hand, running spatiotemporal parameters can also be related to performance in highly trained and elite distance runners (Moore, 2016). They are typically characterized by a more reduced vertical oscillation (Moore, Jones & Dixon, 2014), longer stride (Cavanagh & Williams, 1982) and shorter ground contact time than those displayed by runners of lesser performance level (Mooses et al., 2021). Whereas endurance athletes need to develop metabolic-related abilities such as V̇O2max, distance runners also need to be able to apply great forces onto the ground in short times to achieve high running speeds such as MAS. The latter ability is not a requirement of swimmers, cyclists, or rowers. The foot only can apply force onto the ground to displace the body forward during the running ground contact phase. Runners can reach faster speeds through the application of greater forces onto the ground rather than moving the legs at a faster rate (Weyand et al., 2000). In addition, the increase in running speed demands the application of more force during the same contact time or even shorter (de Ruiter et al., 2016) and this is the reason why running speed should be the same when comparing contact times and force application magnitudes among different runners. Stride frequency and length depend on each other and determine running speed. A greater ground reaction force on its vertical and horizontal components to achieve a faster speed while keeping the contact time constant or shorter would result in either an increased stride length, frequency or both. Interestingly, high correlation values have been found between 100 m performance and that in 5000 m and 10,000 m in elite long-distance runners (Yamanaka et al., 2020), therefore showing how the ability to apply high levels of ground reaction force on its vertical and horizontal components to achieve high speed is related to running performance. In this way, while V̇O2max can be improved using any kind of endurance training (i.e., running, swimming, or cycling) (Rosenblat, Granata & Thomas, 2022), running MAS improvement requires of specific running training methods to be developed (González-Mohíno et al., 2016).

By contrast, it was reported in well-trained subjects that adding alternative training as a complement to running can improve running performance, although not to the same extent as when achieving the same training load through exclusively running (Foster et al., 1995). However, whether middle- and long-distance runners reach different levels of MAS compared with other endurance athletes (i.e., non-runners) with similar V̇O2max has not been previously studied. Understanding the mechanisms by which a greater running performance is achieved through a similar physiological ability (i.e., V̇O2max), which in turn may be related to the behavior of spatiotemporal parameters, may be useful for coaches and practitioners to optimize their running training methods.

Therefore, the aims of this study were: (1) to compare MAS, V̇O2max and maximum capillary blood lactate concentration achieved during an incremental track running test between a group of runners and other of non-running endurance athletes; and (2) to compare running spatiotemporal parameters during the test between both groups, in order to establish an eventual relationship between mechanical efficiency indicators and the achievement of MAS. We hypothesized that V̇O2max would not differentiate the runners from non-runners given that both groups were previously exposed to endurance training which represents a suitable stimulus to improve V̇O2max. Furthermore, we expected that runners displayed a faster MAS than non-runners also explained by their greater ability to deliberately modify their running spatiotemporal patterns as long as the speed increases due to the lack of specific running adaptations in non-runners. Therefore, their MAS, and not their V̇O2max, would be the determining factor of their greater running performance.

Material and Methods

Participants

Twelve male athletes of National level participated in the study: six middle- and long-distance runners (runners), and six endurance non-running athletes (non-runners: cycling, n = 1; orienteering, n = 1; triathlon, n = 2, rowing, n = 1; canoeing, n = 1). Their characteristics are summarized in Table 1. Furthermore, regarding the track performance of the six runners, one athlete had completed the 5,000 m and 10,000 m in 14:20 and 29:30 (min:s), respectively, three other athletes had covered the 1,500 m in 3:43.56, 3:45.48 and 3:54.89 (min:s), respectively, and other runners had run the 800 m in 1:50.32 and 1:55.24 (min.s), respectively, at recent competitions. The sample size was calculated to be a minimum of six athletes per group (see Statistical analysis).

Athletes were informed of the aims and characteristics of the study and signed their prior consent to participate according to the Declaration of Helsinki. The study was approved by the Ethics Committee for Clinical Research of the Sports Administration of Catalonia (EICAEC 10022012).

Procedures

The participants completed an incremental protocol to determine their MAS.

After a standard warm-up (10-min run at 60% of MAS and 5 min of dynamic stretching exercises), under an air temperature of 16−18 °C, relative humidity of 85–90%, and wind velocity <2 m s−1 measured through GAO-ANEMO-101 (Gaotek, USA), participants performed the Université de Montréal track test (UMTT) (Leger & Boucher, 1980) to assess their individual MAS. The UMTT started at 7 km h−1, and speed was increased by 2 km h−1 every minute up to volitional exhaustion or inability to maintain the paced speed (García-Pinillos et al., 2019). The test was performed on a synthetic 400 m athletics track and paced by an assistant riding a bicycle equipped with a calibrated digital speedometer.

Table 1 Characteristics of the two groups of participants.

Data are mean (standard deviation).

 	Age (years)	Height (cm)	Body mass (kg)	
Runners (n = 6)	26.5  (4.0)	176.0 (3.6)	66.7 (0.8)	
Non-runners (n = 6)	23.8 (1.7)	180.5 (4.1)	73.1 (4.8)	
Differences (p)	0.145	0.061	0.007	

Before, during, and after the test, cardiorespiratory and pulmonary gas exchange parameters were telemetrically recorded using a breath-by-breath portable gas analyzer (K4 b2, Cosmed, Italy). The instrument was calibrated before each test according to the manufacturer’s instructions. Oxygen uptake (V̇O2), heart rate, respiratory quotient values during the last 30 s of the test or when plateau was reached were taken as V̇O2max, HRmax, and RQmax, respectively. Running spatiotemporal parameters were determined using a 5-m contact platform (Ergo-Runner Bosco System, Italy) located before the finish line. Stride frequency (steps per second), stride length (m), contact time (ms), and flight time (ms) were recorded at the last two steps of each 400 m lap. The participants were instructed to not accelerate on the five-meter contact platform area. A coefficient of variation (%CV) was calculated for each spatiotemporal variable in each participant from 8 km h−1 onwards to determine the extent to which participants modified each parameter throughout the test to adapt to its increasing speed until exhaustion.

Capillary blood lactate concentration was measured on 20 µl samples by a photoenzymatic method (Photometer 4020, Hitachi, Japan) using Boehringer Mannheim (Germany) lactate reagents at rest and 1-, 3-, 5-, 7- and 10-min pos t-test. The highest value observed among the six [BLa] assessments per test was recorded.

Statistical analysis

Sample size was calculated based on pilot MAS measurements (t-test for two independent samples: α = 0.05, power 1–β = 0.95, d = 0.5), which yielded a minimum of 6 subjects per group (actual measurements power = 0.98).

All analyses were performed with the Statistical Package for Social Sciences 24.0 program (IBM, Armonk, NY, USA). Data are presented as mean ± standard deviation (SD). Normality of data distribution (Kolmogorov–Smirnov test), variance homogeneity (Levene’s test), and assumption of sphericity (Mauchly’s test) as applicable were checked in all parameters. When the sphericity assumption was violated, the Greenhouse-Geisser correction was used.

Spatiotemporal parameters during the MAS test were compared between groups at two running speeds: 8 km h−1 (V8), and 12 km h−1 (V12) through Student’s t-test for independent samples. Between-stage comparisons were made by repeated measures ANOVA with four intra-subject factors/speeds (i.e., V8, V12, the intermediate speed between V12 and MAS [V12-MAS], and MAS). Partial eta-squared (ηp2) values were calculated to determine effect sizes and interpreted as follows: trivial (<0.1), small (0.01–0.04), moderate (0.04–0.11), large (0.11–0.20), or very large (>0.20). Differences were assessed using the Bonferroni’s post hoc test. %CV was assessed for each spatiotemporal variable in each participant considering the aforementioned four speeds. %CV, MAS, V̇O2max, HRmax, RQmax, and maximum [BLa] were compared between groups through Student’s t-test for independent samples. The effect size was assessed using the d statistic (Cohen, 1988), which was considered either trivial (d < 0.20), small (0.20–0.6), moderate (0.6–1.2), large (1.2–2.0), or very large (2.0–4.0) (Hopkins et al., 2009). The level of significance for all tests was p < 0.05.

Results

Runners achieved a faster MAS (21.0 vs. 18.2 km h−1) than non-runners, although V̇O2max, HRmax, RQmax, and maximum [BLa] was not different between both groups (Table 2). Concerning running spatiotemporal parameters, no differences were found between groups in all parameters at V8 and V12.

Both groups showed a similar pattern in spatiotemporal parameters throughout the test, in which contact time progressively decreased (p < 0.001, ηp2 = 0.9) (Fig. 1A), paralleled by a progressive increase in flight time (p <0.001, η p2 = 0.8) across phases (0.001 <p <0.021), except for the final phase (between V12-MAS and MAS) in which flight time stabilized (Fig. 1B). In turn, stride frequency and stride length (Figs. 1C, 1D, respectively) progressively increased during the test (p <0.001, ηp2 = 0.8 and 0.52, respectively) and across phases (0.001 <p <0.035). The %CV of stride length was significantly (p = 0.015) greater in runners than non-runners (Fig. 1D). The %CV of the rest of spatiotemporal parameters were similar between groups.

Discussion

The main aim of the present study was to elucidate whether a group of trained runners displayed different levels of V̇O2max and MAS than a group of trained non-runners endurance athletes during an incremental test. In agreement with our previous hypothesis, runners achieved a faster MAS (vV̇O2max) than non-runners despite displaying a similar V̇O2max and maximal blood lactate concentration. Runners’ stride length patterns were similar between both groups, although runners were able to modify their stride length to a greater extent than non-runners throughout the test in order to achieve a greater MAS, showing a likely a higher running mechanical efficiency than non-runners.

Therefore, differences between runners and non-runners were not explained by participants’ V̇O2max (or maximum blood lactate concentration) and may be attributed to lower body mass ( Myers & Steudel, 1985), and greater running mechanical efficiency in runners, particularly at fast speeds, in line with previous studies (Billat & Koralsztein, 1996; Fernández-Del-Olmo et al., 2002; Williams & Cavanagh, 1987). The faster MAS of the runners (21.0 vs. 18.2 km h−1) was associated to a more stable pattern of spatiotemporal parameters, characterized by a greater increase in stride length throughout the tests, indicating a likely greater running mechanical efficiency (Tartaruga et al., 2012; Williams & Cavanagh, 1987). The fact that both V̇O2max and maximum blood lactate concentration did not differ between groups highlights the great level of endurance performance in both groups and not only in runners. While V̇O2max is considered one of the most important physiological performance determinants in endurance sports (Foster, 1983), the correlation between blood lactate concentration and level of performance in endurance events is very strong (Maffulli, Capasso & Lancia, 1991; Roecker et al., 1998). However, athletes need to develop other specific sport-related (i.e., running) abilities such as those involving the neuromuscular system to be able to achieve high running speeds.

Table 2 Maximum aerobic speed (MAS), V̇O2max, maximum heart rate (HRmax), respiratory quotient (QRmax) and Maximum blood lactate concentration [BLa], in runners and non-runners.

Mean (SD), significance of the differences (p), and group effect size (d) are displayed.

 	Runners	Non-runners	Significance	Effect size	
 			(p)	(d)	
MAS (km h−1)	21.0 (0.6)	18.2 (1.2)*	0.0001	3.0	
V̇O2max (ml kg−1 min−1)	71.6 (5.4)	78.3 (8.8)	0.14	0.9	
HRmax (beats min−1)	194 (3)	189 (12)	0.33	0.5	
RQmax	1.07 (0.2)	1.03 (0.4)	0.55	0.1	
Maximum [BLa] (mmol l−1)	13.6 (1.5)	12.3 (3.7)	0.42	0.5	
Notes.

* Differences between R and NR groups (p < 0.05).

Figure 1 Evolution of kinematic parameters during the maximum aerobic speed (MAS) test for both groups: contact time (A), flight time (B), stride frequency (C), and stride length (D) for each running speed.

An asterisk (*) indicates significant difference (p < 0.05) between runners and non-runners in %CV (coefficient of variation). V8: 8 km h−1; V12: 12 km h−1; V12-MAS: intermediate speed between 12 km h−1 and MAS.

Williams & Cavanagh (1987) reported that 54% of the inter-individual variability existing in running economy could be explained by spatiotemporal variables and demonstrated that most economical runners possess a typical running style. Accordingly, running economy (i.e., the cost of running determined through O2 consumption at a submaximal steady-state intensity) is considered one of the most important physiological performance determinants in distance running (Foster & Lucia, 2007). Interestingly, in a one-case study, Jones (2006) reported that the 8% improvement observed in 3000-m performance in an Olympic runner over five years was achieved with a concomitant 10% decrease in V̇O2max but was associated with MAS improvement. Thus, running performance could be partially explained by mechanical efficiency and capacity to generate greater muscle power (i.e., longer stride length in less time) in a situation of fatigue. Furthermore, adaptations resulting from an increase in running speed also involve specific changes in spatiotemporal parameters such as the observed progressive increase in stride rate, stride length, and flight time, and the progressive decrease in contact time (Fig. 1). Our results are in line with previous investigations focused on the evolution of spatiotemporal parameters during incremental treadmill running tests (Brughelli, Cronin & Chaouachi, 2011; Castro et al., 2013).

On the other hand, it has been shown that endurance training decreases the variability in stride frequency, leading to reduced mechanical and energy cost of running (Slawinski & Billat, 2004). These authors assessed these two main factors according to the training status of three groups of athletes (highly, well, and less trained endurance runners). They reported that highly trained runners did not display a lower energy cost of running than runners of lower training status displayed, but a lower mechanical cost of running. Highly trained runners achieved this more efficient running pattern through a reduction in the amplitude of movement of the center of mass. They concluded that running performance might be associated with the same self-optimizing mechanism contributing to a reduction in the impact loads generated during the initial portion of the support phase of the stride (Slawinski & Billat, 2004).

The greater variation in stride length adopted by runners than non-runners while not decreasing contact time may be then explained by their more efficient ability to produce a wide range in both vertical and horizontal components of ground reaction forces during a constant contact time. Furthermore, stride frequency did not increase in runners to a greater extent than in non-runners to achieve a faster MAS. It has been reported that the ability to select an optimal stride length or stride frequency depends on the runner’s performance level (de Ruiter et al., 2014). This optimal running pattern of spatiotemporal parameters which is achieved through a self-optimization process can be assessed through the acute manipulation of either stride frequency or length so that a curve is derived mathematically to set their most economical levels (de Ruiter et al., 2014). Typically, the optimal stride length is within a range between 3% longer to shorter than the preferred one (Connick & Li, 2014; de Ruiter et al., 2014) and variations of this magnitude cannot modify the running economy (Moore, Jones & Dixon, 2014). However, greater variations than 6% in either stride length or frequency would incur in a further deterioration of running economy (de Ruiter et al., 2014). In this sense, novice runners reported a greater variation in their preferred stride frequency from their optimal stride frequency (i.e., 8%) than that in experienced runners (i.e., 3%) (de Ruiter et al., 2014), therefore showing the greater ability of the latter to adequately self-select their preferred stride frequency. These findings are consistent with those of the present study given that the group of runners were able to increase their stride length to properly suit the increasing speed demands of the incremental test and finally achieved a substantial faster MAS than the group of non-runners. Despite kinetic factors were not analyzed in the present study, the greater variation in stride length and faster MAS in runners than non-runners while variation in contact and flight times and stride frequency throughout the tests, and all spatiotemporal variables at 8 and 12 km h−1, and V̇O2max, remained similar among groups, demonstrate that runners may possess a greater ability to apply force onto the ground than non-runners. According to the synergistic approach to understand the metabolic costs associated to running developed by Arellano & Kram (2014), both vertical and horizontal forward forces which are responsible for supporting body weight and accelerating the body, respectively, are the cause for most of the running metabolic cost. In this way, Støren, Helgerud & Hoff (2011) reported that the sum of both peak vertical and anterior-posterior forces highly inversely correlated with running economy and 3 km performance in elite distance runners. Therefore, experienced and trained runners and more specifically those belonging to the runners’ group of the present study may have learnt to decrease the forces applied onto the ground while maintaining running speed. For example, a lower braking was associated to a greater running economy (Lieberman et al., 2015). In addition, the spring-mass model has been proposed to understand the body’s bounce during the ground contact phase, given that the magnitude of the ground reaction forces in running are proportional to the body’s vertical displacement as if the leg acted as a spring during the contact time (Cavagna et al., 1988; Dalleau et al., 1998; Moore, 2016). Within this context, the concept of stiffness is related to the degree of deformation (i.e., vertical displacement) of the whole body or a part of it (e.g., legs) in relation to the vertical ground reaction force (Butler, Crowell & Davis, 2003; Divert et al., 2005). Leg stiffness highly correlates with running economy (Dalleau et al., 1998) and is related to the ability in runners to apply greater ground reaction forces in shorter contact times (Morin et al., 2007). Therefore, runners might have increased their stride length to a greater extent than non-runners due to their supposedly greater leg stiffness. All the aforementioned abilities and adaptations leading to a more refined pattern of spatiotemporal parameters can be achieved through specific running training as well as other types of strength-related training (Blagrove, Howatson & Hayes, 2018).

Future studies may focus on the assessment and comparison of kinetic characteristics such as ground reaction forces and leg stiffness among well-trained distance runners and non-running endurance athletes. Some limitations must be acknowledged in the present study First, the lack of running economy analysis, which may have differentiated runners and non-runners and linked those differences to the ones found in spatiotemporal parameters and MAS. Second, the lack of kinetic analysis, which would have explained the causes to a greater extent of the relationship between the greater variation in stride length and faster MAS observed in the group of runners. Finally, a greater sample size would have improved the quality of the statistical analysis.

Conclusions

These results demonstrate that middle- and long-distance runners can achieve a faster MAS compared to endurance athletes from other modalities, despite exhibiting a similar V̇O2max and maximum blood lactate concentration. This superior performance may be associated to a greater mechanical efficiency. Overall, at the range of speeds measured, the group of runners exhibited a greater variation in stride length across the tests leading to a faster MAS, which may be related to a more mechanically efficient pattern of spatiotemporal parameters.

Supplemental Information

Supplemental Information 1 Age, weight, height, MAS, and VO2max, blood lactate concentration data, and kinematic parameters in all participants

Click here for additional data file.

The authors gratefully thank the voluntary participation of the athletes studied in the current investigation.

Additional Information and Declarations

Competing Interests

Author Contributions

Human Ethics

Data Availability

The authors declare there are no competing interests.

Arturo Casado conceived and designed the experiments, performed the experiments, analyzed the data, prepared figures and/or tables, authored or reviewed drafts of the article, and approved the final draft.

José Luis Tuimil conceived and designed the experiments, performed the experiments, analyzed the data, prepared figures and/or tables, authored or reviewed drafts of the article, and approved the final draft.

Xavier Iglesias conceived and designed the experiments, performed the experiments, analyzed the data, prepared figures and/or tables, authored or reviewed drafts of the article, and approved the final draft.

Miguel Fernández-del-Olmo conceived and designed the experiments, performed the experiments, analyzed the data, prepared figures and/or tables, authored or reviewed drafts of the article, and approved the final draft.

Pedro Jiménez-Reyes conceived and designed the experiments, performed the experiments, analyzed the data, prepared figures and/or tables, authored or reviewed drafts of the article, and approved the final draft.

Rafael Martín-Acero conceived and designed the experiments, performed the experiments, analyzed the data, prepared figures and/or tables, authored or reviewed drafts of the article, and approved the final draft.

Ferran A. Rodríguez conceived and designed the experiments, performed the experiments, analyzed the data, prepared figures and/or tables, authored or reviewed drafts of the article, and approved the final draft.

The following information was supplied relating to ethical approvals (i.e., approving body and any reference numbers):

The Sports Administration of Catalonia granted Ethical approval to carry out the study within its facilities (Ethical Application Ref: EICAEC 10022012).

The following information was supplied regarding data availability:

The raw measurements are available in the Supplemental File.

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
