# Peer review of "Maximum aerobic speed, maximum oxygen consumption, and running spatiotemporal parameters during an incremental test among middle- and long-distance runners and endurance non-running athletes"

_PeerJ, doi:10.7717/peerj.14035_

## Round 0.1 · original submission · Major Revisions

Please revise your article in line with the reviewers' comments and write a detailed rebuttal on a line-by-line basis.

Reviewer 1 ·

Basic reporting

Basic reporting
Line 14 – I don’t understand what is in the brackets here (partial eta squared is equal to or greater than 0.52 and 0.8?); you need to rewrite this.
The results section of the abstract is slightly odd in that it includes material that is not particularly important or relevant to the research design (why is CV important?). How did you measure mechanical efficiency? Why is an increase in stride length more mechanically efficient?
Title – rather than “kinematic”, what you have measured is better described as “spatiotemporal”.
Lines 6,7 etc. – it seems bizarre that you have shortened “runners” to “R” and “non-runners” to “NR”. Please use the full words throughout the manuscript.
Lines 9-10 – it would make more sense to shorten the relatively long background so that your methods had more information.
Line 13-15 – this is not a particularly interesting finding, so was there any difference between groups of participants?
Line 28 – I think “prescribe” works better than “program”.
Lines 33-36 – at what speed?
Line 39 – I think “requirement” works better than “request”.
Line 50 – “high levels of ground reaction force” does not make sense. Because force is a vector, you need to be specific about the direction and magnitude. Do you mean peak force and, if so, what importance do you place instead on impulse?
Line 64 – I think “would” works better than “did”.
Lines 64-69 – this sentence is so long as to be incomprehensible.
Line 90 – how did you measure wind speed?
Line 183-187 – there isn’t anything particularly interesting about this finding. It is to be expected that faster speeds are associated with higher stride lengths and frequencies (otherwise, how would they occur?).
Lines 188-241 – I would recommend that you remove all of this material. Most of it is not relevant to your study as it refers to variables that you did not measure. Much of it makes no sense in the context of your study design.
Line 255-256 – you don’t know this based on how you conducted the study.
Table 2 – are you sure about the effect size values? 0.5 does not appear correct for the bottom three variables (although in the case of RQ, this might be because the SDs look odd).

Experimental design

Line 74 – please give an objective guide to these athletes’ abilities (e.g., personal record times)
Regardless of your statistical analysis, six participants per group is still a very small sample.
Line 118 – why did you use only two (slow) speeds? It looks later on like you have also measured on another two occasions per test (?).
Line 124 – I don’t understand this point. Did you calculate CV% for only two steps per speed? If so, it does not make sense to use such a small sample. I am not even sure why you have measured CV% given the research question – what does it tell you? In Figure 1, the CV% are enormous – are these across all four speeds measured? It would not make sense to conduct this analysis in this way. Ultimately, there is no point in including CV% as a variable; please remove it completely.

Validity of the findings

Lines 133-134 – presumably, these values were not different at their MAS, showing that the non-runner group had (in effect) worse scores at the same speed.
Line 156 – as I understand it from what you’ve done, the stride length at MAS is longer because they were running faster. You cannot simply compare stride length at different speeds and then claim the longer stride length in the faster group shows they have a higher running mechanical efficiency. The only true comparisons for these spatiotemporal variables is at 8 and 12 km/h.
Line 166 – this is not what the results tell me. They tell me instead that the non-runners had similar physiological results to the runners at their endpoint, which was earlier and at a slower speed. This means that they are not as able to run fast, which is likely to be related to the specific training adopted by runners. We would similarly expect runners to be not as good on tests that were specific to rowing, swimming, etc.

Additional comments

The aim of this study was to measure differences between trained runners and endurance athletes from other sports. There are some good points in the study, but there are major issues with how the results are interpreted. It is also not clear why some variables have been included. There is too much of an emphasis on the spatiotemporal results and not enough on the physiological ones. Because of how the results are reported (relative to MAS), some misinterpretations occur that detract from the study’s important findings.

Reviewer 2 ·

Basic reporting

Basic reporting
Thanks for the invitation to review this manuscript that describes the differences between maximal aerobic speed, VO2max and running kinematics in runners versus non-runners.

General comments:
The manuscript is well-written and clear for the reader. Only a few suggestions are required.

Experimental design

The aim is well stablished and the methodology appropriate.

Validity of the findings

The study can be replicated and the impact of the results is high.

Additional comments

Minor Comments:
Introduction section: I think that the authors need to connect better the paragraphs until the last one, with the aim of the study. Why is it necessary to compare these variables between runners and non-runners?
Methods section:
Line 86: I don't know if this subtitle is the best option, being called the same as the title of this section. Maybe the authors can change to “procedures”.
Line 87: MAS has been defined in the introduction section. It is necessary to include vVO2max in brackets?
Line 98: Oxygen uptake (VO2), heart rate…
Line 98: During the last 30s of the test or when plateau was reached. There are references for that in the literature.
Line 99-102: Reference for this protocol? It is possible any change in speed near to the 5-m contact platform that it can alter the data? Remember the increase in speed each minute.
Line 101: Stride length or step length? Please revise and confirm.
Line 105: Highest value? There were more than one sample per minute of rest?
Line 119: Please, change “ANOVA-RM” to “repeated measures ANOVA”.
Discussion section:
Line 166-171: Another explanation could be that these parameters (VO2max and blood lactate concentration at rest) are well developed in both groups, but to achieve high running speed the athletes need to develop other systems (neuromuscular), and this aspect is sport dependent.
Line 190-193: Long sentence. Please rephrase for a better understanding. I don’t understand well your idea here.

Reviewer 3 ·

Basic reporting

No comment.

Experimental design

No comment.

Validity of the findings

The study adds value to the existing literature, with its results filling the defined knowledge gap, although it is performed on a small sample, a limitation that the authors themselves clearly acknowledge.

Additional comments

Thank you to the Editor for the invitation to review this manuscript. I have thoroughly read the manuscript, checked the figure and tables, raw data and all the Supplemental files. It seems the authors appropriately addressed all the reviewer's concerns, as described in Response to editor and reviewer, provided in the Supplemental files. The manuscript rationale is now clearer, and the manuscript is well presented and easy to follow.
The authors have provided ethical approval statement, the manuscript meets the journal’s article requirements, identifiable info has been removed from the files, and all the experiments were necessary and ethical.

Minor remarks:
Line 12, Abstract – There is a mistake, the ES (for reported MAS) is not 0.9 but 3.0 (according to Table 2)
Line 88, Material and methods – at 60% of MAS – was it predicted MAS or value based on pilot MAS measurements?
Line 162 – in brackets, erase the initials L.V. before the author’s surname
Line 392 – the reference Fernández-Del-Olmo et al. is not listed with other references in alphabetical order.

---

## Round 0.2 · Minor Revisions

Please provide the minor revision suggested by the reviewers.

Reviewer 1 ·

Basic reporting

This has been improved greatly. I did notice however from your comments to another reviewer that you have used stride length rather than step length because you believe this reflects a difference between walking and running. This is not correct; a stride is a complete single gait cycle, and therefore represents two steps. What you have reported in your study is therefore the values for step length and step frequency - you should consider rephrasing these.

Experimental design

This has been explained better.

Validity of the findings

This is explained better.

Additional comments

Thank you for making most of the recommended changes. Some of the others should be considered, at least for future studies.

Reviewer 2 ·

Basic reporting

The paper can be accepted in the current form.

Experimental design

The paper can be accepted in the current form.

Validity of the findings

The paper can be accepted in the current form.

Additional comments

The paper can be accepted in the current form.

Reviewer 3 ·

Basic reporting

The article meets all the required criteria: it is written clearly, in professional English language; references provide sufficient context, raw data are shared; the article follows a professional structure, results relevant to hypotheses are presented.

Experimental design

The presented research is within the aims and scope of the journal and conforms to the strict methodological requirements.

Validity of the findings

Underlying data have been provided, conclusions are limited to supporting results.

Additional comments

Minor remarks:

1. In Table 2 Delete 'R group' and 'NR group' and leave only ‘Runners’ and ‘Non-runners’ in table heading and legend, as suggested by the reviewers.

2. Blood lactate is abbreviated to [BLa] in the Abstract and Methods, but to Lacmax in line 169, line 178 and in Table 2, so please unify the abbreviation throughout the article.

---

## Round 0.3 · accepted · Accept

Your manuscript is acceptable in its current form.